# Understanding the Structural and Catalytic Properties of Al(IV)-2 Acidic Sites of ZSM-5

**DOI:** 10.3390/ma17122824

**Published:** 2024-06-10

**Authors:** Yan Tong, Li Zhang, Hong Ma, Yi Wang, Xiaolong Liu

**Affiliations:** 1School of Materials, Sun Yat-sen University, Shenzhen 518107, China; tongy9@mail2.sysu.edu.cn (Y.T.); mahong3@mail2.sysu.edu.cn (H.M.); 2Lanzhou Petrochemical Research Center, Petrochemical Research Institute, PetroChina, Lanzhou 730060, China; zhangl3@petrochina.com.cn (L.Z.); wyi1@petrochina.com.cn (Y.W.)

**Keywords:** Brønsted acid sites, ZSM-5, FCC (Fluid Catalytic Cracking), D-HMQC, solid-state NMR

## Abstract

It is crucial to identify the structures of active sites to understand how catalysts function and to use that understanding to develop better catalytic materials. ZSM-5 zeolites with dominant Al(IV)-2 sites have been developed in this work. ^1^H-^27^Al 2D HMQC and 2D ^1^H TQ(DQ)-SQ NMR experiments have been performed to investigate the structural properties of this acidic site. The Al(IV)-2 sites have Brønsted and Lewis acid characteristics. The catalytic performance of Al(IV)-2 sites has been tested by n-dodecane cracking reactions. The catalytic results show that the Brønsted acidic strength of the Al(IV)-2 sites is comparable to that of the Al(IV)-1 sites, but the Al(IV)-2 sites’ Lewis acid characteristics provide extra catalytic activity. We have gained valuable insights into the characteristics of Al(IV)-2 acid sites within these materials.

## 1. Introduction

Due to the large consumption of light olefins and petrochemicals for producing plastics, textiles, drug intermediates, and fine chemicals, there will continue to be a massive demand-supply disparity. Utilizing heavy oil feedstocks to convert them into lighter hydrocarbons to produce refined fuels and petrochemicals is becoming more attractive [1,2]. One of the most used zeolite-based catalysts for producing lighter hydrocarbons from n-dodecane by steam catalytic conversion is ZSM-5 zeolite [3,4]. ZSM-5 was first synthesized in 1965 by Landolt and Argauer [5]. Because of its high selectivity and relatively lengthy cycle, ZSM-5 is one of the most studied zeolites and has gained a great deal of interest in both basic research and industrial applications [6,7]. Interlinked sinusoidal and straight channels comprise the 3D pore structure of the ZSM-5 zeolite, which is greatly wanted in catalysis for providing distinct diffusion paths for reactants and products [8]. The tetrahedrally coordinated framework Al atoms in zeolites create Brønsted acid sites through the charge-balancing function of a proton (Si-OH-Al). These bridging acid sites are located in the cavities with specific shapes and scales and only allow reactants with suitable sizes and geometries to approach. Therefore, Brønsted acid sites in H-form zeolites are essential for their spectacular selectivity. The success of zeolites as heterogeneous catalysts is strongly affected by their acidic characteristics. The structure and distribution of Al species in the catalyst framework are important factors in understanding zeolite structure-activity correlations [9,10,11]. This area lacks knowledge due to the lack of zeolites with controlled Al distribution and precise descriptions of Al distribution. This results from the lack of an experimental method to identify precisely where the aluminum is located within the framework. In zeolite science, characterizing Brønsted acid sites and establishing the relationship between structure and activity are crucial for advancing the utilization of zeolites as catalysts. Davis and his coworkers have conducted extensive research to understand the influence of factors such as the Si/Al ratio, crystallization conditions, and post-synthesis treatments on acid site properties [12,13,14,15,16]. This understanding is essential and a significant step toward enhancing catalyst performance through optimization. Although numerous researches examining the impact of various factors on the catalytic performance of ZSM-5 zeolite are available, studies investigating the structure-property correlation resulting from the acidity are still desired.

Solid-state NMR spectroscopy has developed over the last few decades into a powerful tool for zeolite research as a high-sensitivity characterizing method for the local structure of resonant nuclei. The type, strength, accessibility, and concentration of acid sites can be determined through solid-state NMR spectroscopy [17,18]. Directly investigating hydroxyl protons acting as catalytically active Brønsted acid sites is feasible by ^1^H MAS NMR spectroscopy. There is currently a lot of research in the literature to determine if single active sites, multiple sites, or distribution of active acid sites exist in this commercially and academically important catalyst [19,20,21]. However, it is unlikely for ^1^H MAS NMR spectra of bare acidic zeolites to offer precise details regarding the various types and locations of the Brønsted acid protons. Two-dimensional (2D) correlation experiments are necessary to obtain comprehensive structural information about complicated systems, such as the local configurations of atoms concerning one another. Through spy spin-1/2 nuclei, the sequence of HMQC has been proposed as an effective means [22,23,24]. Since there are only two radio frequency pulses on the quadrupolar channel in this sequence, there is less complexity, which improves the efficiency of magnetization transfer. Due to their high gyromagnetic ratio, protons can be used as spy nuclei to increase sensitivity significantly, but this creates a new issue because of the broad ^1^H line shapes that result from the strong ^1^H-^1^H homonuclear dipolar couplings. Modern advancements in fast MAS technology enable the proton-detected D-HMQC (‘D’ means dipolar). In the current work, ^1^H/{^27^Al} D-HMQC experiments at very fast MAS were performed to study ^1^H-^27^Al heteronuclear correlation in ZSM-5 zeolites.

Through applying two-dimensional ^27^Al{^1^H} correlation techniques coupled with a series-connected hybrid (SCH) magnet at 35.2 T, acidic ZSM-5 catalysts can have at least two framework Al(IV) sites with hydroxyl groups [11,25,26]. In addition to the known Al(IV) at the framework bridging acid site (BAS), the second tetrahedrally coordinated Al site (denoted Al(IV)-2) experiences an increased chemical shift relative to the BAS. The second site arises from the partially bonded framework (SiO)4−n-Al(OH)n species, significantly improving catalytic reactivity in benzene hydride-transfer and n-hexane cracking processes. For extending zeolite catalysts to more complex feedstocks, this new structure information is paramount to understanding the function of BAS in HZSM-5 catalysts. They were created either by framework crystallization flaws or insufficient post-synthetic hydrolysis of a framework Al before extra-framework Al was created. The Al siting in the zeolite frameworks was demonstrated to depend on the hydrothermal synthesis conditions rather than being random or regulated by simple rules [27,28]. It is well-known that the strength and density of the acid sites in ZSM-5 can indeed be adjusted or tuned through various synthesis conditions. For example, the Si/Al ratio in the synthesis gel affects the acidity of ZSM-5 and the choice and concentration of mineralizing agents (e.g., sodium hydroxide, tetrapropylammonium hydroxide) in the synthesis gel affect the pH and composition of the gel, which in turn influence the formation of ZSM-5 crystals and the acidity of the resulting material. Therefore, in our work, three different synthesis methods have been used to prepare ZSM-5 zeolites for identifying the acid sites in ZSM-5 [29,30,31]. In the present work, we presented an approach to obtain ZSM-5 with predominate Al(IV)-2 sites and found out that the Al(IV)-2 site is affected by the presence of TPA^+^ in the synthesis gel. Z-I was synthesized using Na cations and ZSM-5 seeds without TPA^+^ cations. TPA^+^ cations were used as OSDAs to synthesize high silica ZSM-5 and nano ZSM-5; neither of these materials’ frameworks contain Al(IV)-2 sites.

## 2. Experimental Section

### 2.1. Zeolite Syntheses

ZSM-5 zeolites with dominant Al(IV)-2 acid sites: The preparation of a synthesis gel involved adding Na_2_SiO_4_·18H_2_O and Al_2_(SO_4_)_2_ to an aqueous solution, followed by the addition of ZSM-5 seed crystals. The solution was then stirred at room temperature for one hour. The molar composition of the synthesis gel was Al_2_O_3_:45SiO_2_:4.5Na_2_O:730H_2_O. The resulting dense gel was placed in 100 mL Teflon-lined stainless-steel autoclaves and statically heated at 185 °C and under autogenous pressure for 32 h. The solid products were obtained through filtration, washed with water and ethanol, and dried at 100 °C overnight. These ZSM-5 zeolites were labeled as Z-I.

High silica ZSM-5 zeolites: An aqueous solution containing tetrapropylammonium hydroxide was mixed with Na_2_SiO_4_ and Al_2_(SO_4_)_2_ in a molar ratio (Si/Al) of 550. Then, 1M H_2_SO_4_ was added to attain a pH of 7–8. The synthesis reactions took place in stainless-steel reactors equipped with Teflon cups and were conducted at 180 °C for 32 h. These resulting ZSM-5 zeolites were labeled as Z-II.

Nano ZSM-5 zeolites: The first step is preparing the solution with seeds. The synthesis involved preparing mixtures with a molar composition expressed in terms of oxide ratios (10 (TPA)_2_O:60SiO_2_: 0.5Al_2_O_3_:936H_2_O). This was achieved by adding aluminum sulfate, sodium silicate, and tetrapropyl-ammonium hydroxide in water, stirring for 10 min. The clear solutions were poured into 100 mL Teflon-lined stainless-steel autoclaves and heated statically at 120 °C for 24 h, forming 60 mL of gel. In the second step, new synthesis mixtures were created by adding NaAlO_2_ and Na_2_SiO_4_ to an aqueous sodium hydroxide solution. The molar composition of the synthesis gel was Al_2_O_3_:45SiO_2_:4.5Na_2_O:730H_2_O. The solution was stirred for one hour at room temperature, after which 5 mL of the gel with seed crystals was added dropwise. The resulting gel was placed in 100 mL Teflon-lined stainless-steel autoclaves and heated statically at 120 °C for 32 h. The solid products were obtained through centrifugation, washed with water and ethanol, and dried at 100 °C overnight. These resulting ZSM-5 zeolites were designated as Z-III.

Z-I and Z-II were ion-exchanged with a 1 M NH_4_Cl solution 3 times at 80 °C. Then, the solid materials were filtered out, washed with deionized water, and dried at 100 °C. HZSM-5 was obtained by calcination in air at 823 K for 2 h after heating at a 2 K/min rate. However, Z-III is not necessary in these cases, especially when organic TPA acts as the SDAs; in such cases, HZSM-5 can be directly formed after calcination in the air at 823 K.

### 2.2. Catalyst Characterization

A Bruker Avance 600 NMR spectrometer (Sun Yat-sen University, Guangzhou, China) was used to record nuclear magnetic resonance (NMR) spectra. A typical Bruker MAS probe head with a rotor diameter of 1.9 mm and a spinning frequency of 25 kHz was employed for the ^27^Al magic angle spinning (MAS) NMR. Using a single 1 µs excitation pulse and a 1 s interscan delay, the ^27^Al chemical shift was referenced to a saturated Al(NO_3_)_3_ solution. ^1^H/{^27^Al} D-HMQC experiments were performed under a MAS frequency of 25 kHz and a recycle delay of 1.5 s. For ^1^H, the lengths of π/2 and π pulses were 0.9 and 1.8 µs, respectively. SR4_1_^2^ was used as the recoupling sequence to reintroduce ^1^H-^27^Al heteronuclear dipolar interaction. The ^1^H rf field strength during SR4_1_^2^ was adjusted to 50 kHz, two times the spinning frequency of 25 kHz, and the total recoupling duration was optimized. For ^27^Al, the rf field strength was calibrated using 1 M Al(NO_3_)_3_ aqueous solution. The rf field strength for hard pulse was 250 kHz, while it was set to 1.5 kHz for soft pulse.

Further, 2D ^1^H DQ(TQ)-SQ NMR experiments were carried out using a 1.9 mm MAS probe at 25 kHz spin rates, typical π/2-pulse lengths of 1.5 μs, and a recycle delay of 5 s. The spectra were referenced concerning tetramethylsilane (TMS) using solid tetrakis(trimethylsilyl)silane as a secondary standard (0.27 ppm for ^1^H). The symmetry pulses were used for DQ(TQ) excitation/reconversion, which has been demonstrated to be an efficient homonuclear DQ recoupling scheme at fast MAS rates.

All NMR spectra are presented in the Appendix A.

X-ray powder diffraction (XRD): XRD patterns were measured with a Rigaku Ultimate VI X-ray diffractometer (40 kV, 40 mA) using Cu-Kα1 (λ = 1.5406 Å) radiation (Wilmington, MA, USA).

Scanning electron microscopy (SEM): SEM measurements were carried out on a Hitachi Regulus SU8230 FE-SEM (Tokyo, Japan).

### 2.3. Catalytic Activity Measurements

The zeolites were hydrothermally deactivated for 4 h at 800 °C with 100% steam before the test, respectively. Then, the activity of the zeolite was evaluated by using a microactivity test (MAT) unit (MAT-Ⅱ), which was designed according to the ASTM D-3907 method using a dodecane (analytical purity). The tests were carried out at a reaction temperature of 460 °C, a reacting time of 70 s, and a catalyst-to-oil ratio of 3.2:1. The liquid product was analyzed by Varian 3800 (Palo Alto, CA, USA), and the average cracking activity of the zeolite was obtained. The quality of a fuel’s ignition is measured by its octane number. Two methods for determining the octane number are the research octane number (RON) and the motor octane number (MON). In this study, the RON and MON of the liquid product were determined. The composition of the liquid product and its octane number were analyzed using an Agilent 8890 instrument (Santa Clara, CA, USA).

## 3. Results and Discussion

The XRD powder patterns and SEM photographs of ZSM-5 samples are shown in Figure 1. The X-ray diffraction pattern shows the crystalline samples with reflections that agree with the pattern of standard ZSM-5 zeolite, and the SEM photographs of Z-I and Z-II are coffin-shaped particles. The composition of the synthesis solution, including the types and concentrations of alkali metals, fluoride ions, organic structure-directing agents (OSDAs), or seeding agents, can significantly impact the morphology and properties of ZSM-5. In this work, three popular hydrothermal synthesis methods have been employed to prepare ZSM-5 zeolite, and XRD patterns and morphologies of ZSM-5 reflect the synthesized conditions. Figure 2 shows the ^1^H and ^27^Al MAS NMR spectra of ZSM-5 samples. Modern solid-state NMR spectrometers can distinguish the various types of hydroxyl groups by their chemical shifts due to the resolution of their ^1^H MAS NMR spectra. Signals between 1.2 and 2.2 ppm point to the presence of SiOH groups, for example, on the outside of silicate and aluminosilicate particles or in zeolite framework defects [32,33], i.e., these surface hydroxyl groups can originate from the hydrolysis of silanol groups (Si-OH) located on the external surface or within the channels of the inner surface of the zeolite. The variation in chemical shifts may arise from their location on the external surface or within the zeolite micropores. Due to bridging OH groups in large cages and tiny structural units, the Si-OH-Al group ^1^H MAS NMR signals can be observed at 3.6–4.3 and 4.6–5.2 ppm [34]. Therefore, strongly acidic hydroxy groups in zeolites are characterized by peaks in the region of 3.6–6.0 ppm in the ^1^H MAS NMR. Signals around 4.0–4.5 and 2.5–2.9 ppm in the ^1^H solid-state NMR spectra of dehydrated H-ZSM-5 have been employed as indicators for the presence of hydroxyl groups on bridging acid sites (BAS) and extra-framework aluminum (EFAl) species, respectively. NMR techniques based on heteronuclear dipolar (through space) interactions between ^1^H and ^27^Al nuclei are needed to assign ^1^H chemical shifts to specific bonding situations. A recently developed NMR pulse sequence called D-HMQC is very well suited for examining the spatial proximity of quadrupolar and spin-1/2 nuclei [24]. Figure 3b demonstrates that the tetrahedral Al site is spatially close to three chemically different protons at 2.68, 1.19, and 0.87 ppm. Based on extensive literature reporting known ^1^H chemical shift values [18], the chemical shift at 0.87 ppm and 1.19 ppm can be attributed to the silanol groups on the external surface and the lattice defects, respectively. As shown in Figure 3b, these silanol groups are near framework aluminum species [34]. In the literature, the 2.68 ppm ^1^H signal has been attributed to hydroxyls on extra-framework aluminum species other than tetrahedral Al(IV) species such as Al(III), Al(V), or Al (VI) [18,35]. However, NMR results show that the protons giving rise to the signal at 2.5–2.9 ppm are active Brønsted sites at partially bonded Al(IV)-2 species and Al(IV)-2 sites with a chemical shift at 54 ppm [25]. ^27^Al MAS NMR and D-HMQC data show that Z-I zeolites mainly have Al(IV)-2 sites associated with protons at 2.68 ppm. The broad signal at 5.70 ppm might be generated by disordered bridging OH groups in the zeolites H-ZSM-5, and these hydroxyl protons interact strongly with nearby oxygen atoms in the framework [18]. Therefore, without organic structure-directing agents such as TPA^+^, the framework Al sites in the seeded synthesized ZSM-5 comprise Al(IV)-2 sites. The ZSM-5 zeolites with predominate Al(IV)-2 acid sites offer a solid foundation for research into the characteristics of Al(IV)-2 acid sites in their framework. Our synthesis method shows that framework crystallization defects cause the Al(IV)-2 sites. Chen and his coworkers proposed several intermediate structures formed by Al(IV)-2 acid sites [25]. In this work, the structural properties of defects created by Al(IV)-2 are further studied by 2D multiple quantum ^1^H NMR techniques. Due to the large proton-proton dipolar couplings, fast MAS alone is frequently insufficient to offer high resolution in the crowded regions of ^1^H MAS spectra. ^1^H MAS NMR spectrum of ZSM-5 with dominant Al(IV)-2 acid sites shown in Figure 1a is a good example. It is challenging to resolve the signals from 0 to 5 ppm effectively. Homonuclear ^1^H-^1^H double quantum (DQ) and triple quantum (TQ) 2D NMR experiments have been performed to assign the other signals. The dipolar coupling is strongly dependent on the inter-nuclear distance. It can be used to determine the geometry of the coupled spin, and then 1H TQ/DQ-SQ MAS NMR correlation spectroscopy has been used to identify the SiOH groups and BAS in ZSM-5 [36,37,38]. In 2D NMR spectra, the cross-correlation signal indicates that species with protons are in mutual proximity within about 5 Å (to a maximum of 8 Å). No signal will occur for isolated ^1^H species in such spectra due to the DQ filtering. In the 2D ^1^H DQ(TQ)-SQ spectrum, the spectral span of the DQ (TQ) dimension is twice (or triple) that of the SQ dimension, i.e., the sum of the isotropic chemical shifts of the two (or three) identical or different spins in the SQ dimension yields the chemical shift value of a corresponding contour in the DQ (TQ) dimension. The chemical shift at 0.87 ppm and 1.19 ppm can be attributed to the silanol groups on the external surface and the lattice defects, respectively. According to the 2D ^1^H TQ-SQ NMR spectrum in Figure 4, the contour signals at (2.68, 2.68 + 2.68 + 2.68) suggest that the defects created by Al(IV)-2 acid sites have three OH terminals with the same chemical shifts, as in the hydroxides in Al-(OH)_3_ groups (with structural models such as Figure 4d in Chen’s paper [25]). The chemical shift at 7.00 ppm can be attributed to the bridge-bonded H, which is associated with the framework of a partially hydrolyzed but still bonded Brønsted site. A two-dimensional ^1^H DQ-SQ NMR experiment was performed to reveal the structure formed by Al(IV)-2 acid sites. The correlated contour signals in Figure 5 indicate that the proton with chemical shift at 7.00 ppm is spatially close to the protons with chemical shift at 2.68 ppm. Therefore, the true chemical shifts of Al(IV)-2 acid sites is 7.00 ppm. The signal with 2.68 ppm is from Al-(OH)_3_ terminals. As suggested by the low ^1^H chemical shift value of 2.68 ppm, the protons associated with Al-(OH)_3_ terminals could be weakly acidic. However, the protons associated with Al(IV)-2 acid sites with a high ^1^H chemical shift value of 7.00 ppm are the frame-bonded Brønsted sites. Figure 3b demonstrates that no correlated signal is from ^1^H at 7.00 ppm, which indicts that the ^1^H/^27^Al D-HMQC technique is not reliable for studying the Al(IV)-2 Brønsted sites for the rapid proton chemical exchange. ^1^H/^27^Al D-HMQC experiments at very fast MAS have also been performed to study ZSM-5 (high silica) and ZSM-5 (nano), but no signals have been obtained. The reason is that the protons associated with the Al(IV)-1 site are delocalized through rapid chemical exchange for the clusters of water molecules around the Al(IV)-1 site. The ^1^H/^27^Al D-HMQC technique is not reliable for studying both the Al(IV)-1 Brønsted sites and the Al(IV)-2 Brønsted sites for the rapid proton chemical exchange. Generally, in ^27^Al magic angle spinning (MAS), NMR spectra of zeolites’ approximate chemical shifts are at 50–60 ppm for tetrahedral oxygen-coordinated aluminum, at 30–50 ppm for tetrahedral non-framework Al species, at 30 ppm for five-fold coordinated species, and at 0 ppm for octahedral aluminum [39]. The chemical shifts of ZSM-5 (high silica) and ZSM-5 (nano) are 55 ppm and 53 ppm, respectively. Therefore, we can suggest that both ZSM-5 (high silica) and ZSM-5 (nano) have only Al(IV)-1 acid sites. These ‘second’ Brønsted sites significantly increase catalyst reactivity in benzene hydride-transfer and n-hexane cracking reactions [25]. Therefore, it is important to note that the protons associated with Al(IV)-2 are not weakly acidic. According to our observation, we can suggest that the strong acidic site is from the proton Brønsted site with 7.00 ppm. The proton’s chemical shift at 2.68 ppm is from aluminol groups, which are weak Lewis acids. This assignment agrees with the literature, in which the 2.68 ppm ^1^H signal has been attributed to hydroxyls on extra-framework aluminum species [18,35]. The structural property of Al(IV)-2 acidic sites of ZSM-5 is depicted in Figure 1, which clearly shows that Al(IV)-2 sites have Brønsted and Lewis acid characteristics. The channel intersections where TPA^+^ cations prefer to be are also the locations of Al(IV)-2 acidic sites. Aluminol groups found it difficult to live alongside TPA^+^ cations because of their size. Therefore, when TPA^+^ cations were employed in the synthesis system, Al(IV)-2 acid sites could not be formed.

In this work, the cracking of n-dodecane was performed to determine the acidic strength of two acid sites and evaluate the acid properties of ZSM-5 zeolites. The microactivity test (MAT) unit was employed to gauge the catalytic efficiency of the synthesized zeolites in the steam catalytic cracking of n-dodecane, and the resulting reaction products were meticulously analyzed. During cracking n-dodecane, Brønsted acid sites play the dominant catalytic roles. According to the catalytic cracking performance of n-dodecane shown in Figure 6, the acidic strength of ZSM-5 with Al(IV)-2 acid sites is not less than the acidic strength of ZSM-5 with Al(IV)-1 acid sites. Lewis acidity in protonic zeolite is due to available coordinative unsaturated Al^3+^ ions. It is often caused by EFAl, which exists in several oxide and hydroxide forms, such as Al(OH)^2+^, Al(OH)_2_^+^, AlOOH, Al_2_O_3_, Al(OH)_3_, or multinuclear clusters [40,41]. Apart from Brønsted acid sites, certain aluminum species in zeolites possess Lewis acidic characteristics, and Lewis acidic aluminum plays a pivotal role in traditional cracking reactions [42,43,44]. Generally, the catalytic cracking of alkanes can initially take place on Brønsted acid sites by forming carbonium ions (i.e., via protolytic cracking) or on Lewis acid sites by forming carbenium ions (i.e., via a classical β-scission mechanism) [45]. Corma et al. conclude that both pathways occur in parallel and protolytic cracking has a lower activation energy than β-scission [46]. The tertiary carbenium ions are more stable in branched compounds produced by isomerization processes. Therefore, Lewis acid is favorable for forming isomerized hydrocarbon. The acidity of ZSM-5 catalysts, primarily due to the presence of Brønsted and Lewis acid sites, plays a crucial role in catalytic reactions. Variations in acidity can impact catalytic activity and selectivity in acid-catalyzed reactions. Brønsted acid sites are characterized by their ability to donate protons to reactant molecules, facilitating acid-catalyzed reactions such as cracking, isomerization, alkylation, and dehydration. The catalytic products suggest that the Brønsted acidic strength of the Al(IV)-2 sites is either a little stronger than or comparable to that of the Al(IV)-1 sites. Lewis acid sites are often associated with aluminum atoms in distorted tetrahedral positions or extra-framework cations. These sites exhibit a strong affinity for electron-rich species. They can catalyze reactions involving Lewis acid-based interactions, such as adsorption, activation, and the transformation of unsaturated molecules (e.g., olefins, aromatics). Lewis acid sites are essential for catalyzing alkene oligomerization, alkyl transfer, and aromatic substitution. Based on the catalytic findings, it is evident that both Z-II and Z-III zeolites possess a greater abundance of Lewis acid sites compared to Z-I zeolites. This abundance enhances their ability to catalyze isomerization reactions, resulting in a higher concentration of iso-paraffins in the liquid product.

Since the 1980s, commercial ZSM-5-based additives have been employed in the FCC process to increase the octane number of gasoline. RON and MON are two characteristics that are widely used in the compression ignition (CI) engine community. It is important to accurately record the octane rating of liquid products to ensure that these studies can further improve the design of fuels. The overall conversion of n-dedicate to produce gasoline reformate of different ZSM-5 zeolites was evaluated by RON and MON and shown in Figure 7. High conversion and steady selectivity were maintained during the FCC reaction, resulting in a product with a highly researched RON and MON. The ZSM-5 with predominant Al(IV)-2 acid sites also have high catalytic activity.

## 4. Conclusions

In this study, the combination of 2D D-HMQC and 2D ^1^H multi-quantum MAS NMR experiments provides valuable information on Al(IV)-2 acid sites: the low ^1^H chemical shift value of 2.68 ppm is from hydroxy groups in Al(OH)_3_ and the high ^1^H chemical shift value of 7.00 ppm corresponds to frame-bonded Brønsted sites. Two chemically distinct Brønsted sites can be manipulated through appropriate adjustments to the synthesis conditions. The catalytic products suggest that the Brønsted acidic strength of the Al(IV)-2 sites is either a little stronger than or comparable to that of the Al(IV)-1 sites. According to the catalytic results, it is clear that both Z-II and Z-III zeolites exhibit a higher concentration of Lewis acid sites than Z-I zeolites. This capability of controlling aluminum distribution holds great potential for tuning the properties of silicon-rich zeolites utilized as catalysts in various redox and acid reactions and investigating the effects of aluminum distribution on these materials’ properties and catalytic performance.

## Data Availability

The original contributions presented in the study are included in the article, further inquiries can be directed to the corresponding author.

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
