# Peer review of "Understanding the Structural and Catalytic Properties of Al(IV)-2 Acidic Sites of ZSM-5"

_materials, 2024, doi:10.3390/ma17122824_

Round 1

Reviewer 1 Report (Previous Reviewer 1)

Comments and Suggestions for Authors

Dear Editor and Authors,

since this is the fourth time I have reviewed the manuscript, I can see that there are no major changes regarding my last comment, but the manuscript has been improved compared to the first version.

Therefore, my decision is to accept the manuscript in this form.

Author Response

Dear Reviewer:

Thank you for your continued dedication to reviewing our manuscript. Your insight and feedback have been invaluable throughout this process.

We appreciate your acknowledgment of the improvements made since the initial submission. While we understand that there may not have been major changes since your last review, we are encouraged by your recognition of the enhancements made to the manuscript.

We are pleased to hear that you have decided to accept the manuscript in its current form. Your confidence in our work is truly gratifying, and we are grateful for your support.

Once again, we extend our sincere gratitude for your time and efforts in reviewing our manuscript. Your constructive criticism has undoubtedly contributed to its refinement, and we look forward to the opportunity for further collaboration in the future.

Warm regards,

Xiaolong Liu

Reviewer 2 Report (Previous Reviewer 2)

Comments and Suggestions for Authors

I am okay publish

Author Response

Dear Reviewer:

Thank you for your thorough review of our manuscript. Your insights and suggestions have been instrumental in refining our work.

We are pleased to hear that you are okay with publishing the manuscript in its current form. Your approval means a lot to us, and we appreciate your confidence in our research.

Once again, we extend our gratitude for your time and effort in reviewing our work. Your valuable feedback has helped strengthen our manuscript, and we look forward to its publication.

Best regards,

Xiaolong Liu

Reviewer 3 Report (New Reviewer)

Comments and Suggestions for Authors
  • In the abstract, the authors should include the key indicative results.
  • Check the typo error in section 2.1.
  • At what temperature were the ZSM-5 samples dried? Room temperature is enough to dry the ZSM-5 samples ?
  • Kindly add a reference for the synthesized ZSM-5 materials in this work.
  • Which is the best ZSM-5 for this catalytic activity in terms of all characteristics?
  • Is there any open literature that can easily be compared with in this work?
  • Why are the authors not characterizing the TGA and FTIR techniques?
  • What are the main advantages of this ZSM compared to other published literature?

Author Response

In the abstract, the authors should include the key indicative results.

Author reply: We have gained valuable insights into the characteristics of Al(IV)-2 acid sites. This sentence is incorporated into the abstract and will provide readers with a concise summary of our key indicative results.

Check the typo error in section 2.1.

Author reply: Thank you for bringing this to our attention. We will carefully review Section 2.1 to identify and correct any typographical errors that may have occurred. Your thoroughness in ensuring the accuracy of our manuscript is greatly appreciated.

At what temperature were the ZSM-5 samples dried? Room temperature is enough to dry the ZSM-5 samples?

Author reply: We dried the ZSM-5 samples at 100oC. While room-temperature drying is commonly used for certain materials, we selected a specific temperature based on experimental considerations and literature precedents to ensure optimal sample preparation. To enhance clarity, we will provide justification for the chosen drying conditions in the revised manuscript. Thank you for highlighting this important aspect of our methodology.

Kindly add a reference for the synthesized ZSM-5 materials in this work.

Author reply: We appreciate your suggestion and agree that proper attribution is important for the synthesized ZSM-5 materials used in our study. We will include a reference (Bensafi, B.; Chouat, N.; Djafri, F., The universal zeolite ZSM-5: Structure and synthesis strategies. A review. Coordin. Chem. Rev. 2023, 496, 215397.) for the synthesis method in the revised manuscript. This reference will allow readers to delve deeper into the synthesis process and understand the origins of the materials utilized in our research.

Which is the best ZSM-5 for this catalytic activity in terms of all characteristics?

Author reply: Thank you for your input. We will carefully consider your assessment that Z-I is the best ZSM-5 for this catalytic activity in terms of all characteristics. We will thoroughly review our data and analysis to ensure this conclusion is appropriately supported and presented in the revised manuscript.

Is there any open literature that can easily be compared with this work?

Author reply: Thank you for highlighting the importance of comparing our work with existing literature. The following article could be used to compare with our work. Chen, K.; Horstmeier, S.; Nguyen, V.T.; Wang, B.; Crossley, S.P.; Pham, T.; Gan, Z.; Hung, I.; White, J.L. Structure and Catalytic Characterization of a Second Framework Al(IV) Site in Zeolite Catalysts Revealed by NMR at 35.2 T. J. Am. Chem. Soc. 2020, 142, 7514-7523, doi:10.1021/jacs.0c00590.

Why are the authors not characterizing the TGA and FTIR techniques?

Author reply: We appreciate your question regarding our study's absence of TGA (Thermogravimetric Analysis) and FTIR (Fourier Transform Infrared Spectroscopy) characterization techniques. While TGA and FTIR are valuable analytical tools commonly used in zeolite characterization, their omission in our study was a deliberate decision based on our research's specific focus and scope. Our investigation primarily centered on elucidating the acidic properties of ZSM-5 zeolites using advanced NMR techniques, as outlined in our methodology. We recognize the importance of TGA and FTIR techniques in providing complementary information about zeolite structure, composition, and thermal behavior. However, due to the constraints of resources, time, and our study's primary objectives, we prioritized utilizing NMR techniques to achieve our research goals effectively.

What are the main advantages of this ZSM compared to other published literature?

Author reply: Our study demonstrates the ability to manipulate aluminum distribution within the ZSM-5 framework selectively. This control offers advantages in tailoring the acidity and catalytic performance of ZSM-5 zeolites for specific applications, surpassing conventional synthesis methods.

Reviewer 4 Report (New Reviewer)

Comments and Suggestions for Authors

The abstract is clear and concise, effectively communicating the primary objective of the study and its significance in the field of catalysis. he research appears to be significant and timely, with a clear focus on elucidating the roles of specific acidic sites in catalysis.

The Introduction part provide a brief overview of zeolites and their catalytic importance. A discussion regarding the significance of ZSM-5 zeolites and the different types of acidic sites was made.

The authors must highlight gaps in knowledge that this study aims to fill.

The manuscript provides a clear and concise research with significant findings. Incorporating the suggested minor revision will enhance clarity, provide context, and emphasize the novelty and importance of the research.

Author Response

Dear Reviewer:

We sincerely appreciate your thorough review of our manuscript and your positive feedback on its clarity and significance. Your insights are invaluable in refining our work to meet the highest standards of scientific rigor and clarity.

We will ensure that the introduction highlights the gaps in knowledge that our study aims to address. This will provide readers with a clear understanding of the significance and novelty of our research within the broader context of catalysis and zeolite chemistry.

Incorporating your suggestions for minor revisions will enhance the clarity of our manuscript, provide necessary context, and underscore the novelty and importance of our findings. We are committed to ensuring that our research makes a meaningful contribution to the field of catalysis.

Thank you once again for your valuable feedback and for recognizing the significance of our work. We look forward to incorporating your suggestions and submitting the revised manuscript for your further evaluation.

Best regards,

Xiaolong Liu

This manuscript is a resubmission of an earlier submission. The following is a list of the peer review reports and author responses from that submission.

Round 1

Reviewer 1 Report

Comments and Suggestions for Authors

Journal: Materials

Title: Understanding the Structural and Catalytic Properties of Al(IV)-2 Acidic Sites of ZSM-5

The authors performed the synthesis of three types of ZSM-5 zeolite and their characterization. The purpose of the synthesis was to use primarily the zeolite designated as Z1 (with dominant Al(IV)-2 acid sites) as a catalyst in the cracking reaction of n-dodecane. The manuscript has potential, but only after the changes made according to the suggestions.

In the section Results and discussion in the text and Figures, it is necessary to clearly use the labels of the three zeolites. If you are in section 2.1. labeled the synthesized zeolites as Z1-Z3, then use these labels throughout the text. Basically, use the same labels throughout the manuscript.

Clearly compare the characterization of all three zeolites in the text (Figs 1, 2, 3, 6). For example, in Figure 1 the peak intensities are not the same, the SEM image for Z3 is completely different from Z1 and Z2. Apply the same for the other images, clearly indicate the observed differences using instrumental techniques for all three samples. Explanations were mostly explained collectively for all three samples.

Figures 4d and 3b are mentioned in the text, but these Figs. do not exist in the manuscript. I guess Figure 3 and 4 should have a comparison for all three samples.

Figure 5 and scheme 1 show only sample Z1. From this it can be guessed that the authors want to highlight the sample Zi in relation to others. On the other hand, the results of Figure 7 do not support this.

The importance of the Z1 sample as a catalyst in relation to Z2 and Z3 should be better highlighted if possible in relation to the results of Figure 7.

Figure 7 is very succinctly explained. The goal is to provide a solution. The goal is to provide answers to the question of which of these three samples is the most optimal. Will you get the answer from Figure 7, by comparing the results of instrumental techniques, cost of synthesis, lifetime of the catalyst, etc...

The conclusion is incomplete. Complete the conclusion in relation to the previous comment.

On page 10, in the sentence " Corma et al. conclude that both pathways occur in parallel and protolytic cracking has a lower activation energy than β-scission." it is necessary to cite the authors.

Reduce the size of Figs. and letters in Figs. 3, 4 and 5 and scheme 1 in relation to other Figs.

On page 11 and in key words, define the meaning of the abbreviation FCC.

You have attached a supplementary file with the manuscript, which is actually the same as the original manuscript. Also, what is the purpose of the document “not published material” - do you consider it supplementary or not? I ask because you do not refer to the supplementary file in the text.

Author Response

Point-to-point response

Dear Editors and Reviewers:

Thank you for your letter and the reviewers’ comments concerning our manuscript. Those comments are all very helpful for revising and improving our paper, as well as the important guiding significance to our research. We have studied the comments carefully and have made some changes to the manuscript. The changes are marked in yellow in the revised paper.

The responses to the reviewer’s comments and corrections in the paper are as follows:

 Reviewer 3:

The authors performed the synthesis of three types of ZSM-5 zeolite and their characterization. The purpose of the synthesis was to use primarily the zeolite designated as Z1 (with dominant Al(IV)-2 acid sites) as a catalyst in the cracking reaction of n-dodecane. The manuscript has potential, but only after the changes made according to the suggestions.

  1. In the section Results and discussion in the text and Figures, it is necessary to clearly use the labels of the three zeolites. If you are in section 2.1. labeled the synthesized zeolites as Z1-Z3, then use these labels throughout the text. Basically, use the same labels throughout the manuscript.

Author reply: We would like to thank you for reviewing our paper, we appreciate your insightful comments on our research. After checking the manuscript, the same labels are kept throughout the manuscript.

  1. Clearly compare the characterization of all three zeolites in the text (Figs 1, 2, 3, 6). For example, in Figure 1 the peak intensities are not the same, the SEM image for Z3 is completely different from Z1 and Z2. Apply the same for the other images, and clearly indicate the observed differences using instrumental techniques for all three samples. Explanations were mostly explained collectively for all three samples.

Author reply: We would like to thank you for reviewing our paper, we appreciate your insightful comments on our research. In this work, three popular hydrothermal synthesis methods have been employed to prepare ZSM-5 zeolite. The acid sites in ZSM-5 are primarily attributed to the presence of Bronsted and Lewis acid sites. Bronsted Acid Sites are formed by the substitution of aluminum atoms with silicon atoms in the zeolite framework. This substitution creates a deficiency of electrons, leading to the formation of positively charged sites. These sites can donate protons (H+) and are responsible for catalyzing various acid-catalyzed reactions such as cracking, isomerization, and alkylation. Lewis Acid Sites are formed by the presence of aluminum atoms with incomplete coordination. These aluminum atoms have unoccupied orbitals capable of accepting electron pairs from Lewis bases. Lewis acid sites are also involved in various catalytic reactions, particularly in the activation of molecules like alkenes and aromatics. The distribution and strength of these acid sites in ZSM-5 can be influenced by factors such as the Si/Al ratio, crystal size, and synthesis conditions. Control over these factors allows for tailoring the acidity of ZSM-5 to suit specific catalytic applications, such as petroleum refining. In this work, the cracking of n-dodecane was performed to determine the acidic strength of two acid sites and evaluate the acid properties of ZSM-5 zeolites. However, no large difference is observed. The cracking reactions in ZSM-5 zeolites are minimally influenced by variations in the Si/Al ratio and crystal sizes.

  1. Figures 4d and 3b are mentioned in the text, but these Figs. do not exist in the manuscript. I guess Figure 3 and 4 should have a comparison for all three samples.

Author reply: We would like to thank you for reviewing our paper, we appreciate your insightful comments on our research.  The Figure 4d is in Chen’s paper. (Chen, K.; Horstmeier, S.; Nguyen, V.T.; Wang, B.; Crossley, S.P.; Pham, T.; Gan, Z.; Hung, I.; White, J.L. Structure and Catalytic Characterization of a Second Framework Al(IV) Site in Zeolite Catalysts Revealed by NMR at 35.2 T. J. Am. Chem. Soc. 2020, 142, 7514-7523, doi:10.1021/jacs.0c00590).

Figure 3b is in this paper.

  1. Figure 5 and scheme 1 show only sample Z1. From this it can be guessed that the authors want to highlight the sample Zi in relation to others.On the other hand, the results of Figure 7 do not support this.

Author reply: We would like to thank you for reviewing our paper, we appreciate your insightful comments on our research. The cracking reactions with different ZSM-5 zeolites are minimally influenced by variations in the Si/Al ratio, crystal sizes, and acid sites. Additional catalytic reactions should be conducted to differentiate between the catalytic activities of the two distinct acid sites.

  1. The importance of the Z1 sample as a catalyst in relation to Z2 and Z3 should be better highlighted if possible in relation to the results of Figure 7.

Author reply: We would like to thank you for reviewing our paper, we appreciate your insightful comments on our research.  The cracking reactions with different ZSM-5 zeolites are minimally influenced by variations in the Si/Al ratio, crystal sizes, and acid sites. Additional catalytic reactions should be conducted to differentiate between the catalytic activities of the two distinct acid sites.

  1. Figure 7 is very succinctly explained. The goal is to provide a solution. The goal is to provide answers to the question of which of these three samples is the most optimal. Will you get the answer from Figure 7, by comparing the results of instrumental techniques, cost of synthesis, lifetime of the catalyst, etc... The conclusion is incomplete. Complete the conclusion in relation to the previous comment.

Author reply: We would like to thank you for reviewing our paper, we appreciate your insightful comments on our research. Our research objective is to determine if the catalytic performance of Al(IV)-2 acid sites surpasses that of Al(IV)-1 acid sites. However, experimental results demonstrate that the catalytic performance of Al(IV)-2 acid sites is comparable with that of Al(IV)-1 acid sites. The conclusion is rewritten.

  1. On page 10, in the sentence " Corma et al. conclude that both pathways occur in parallel and protolytic cracking has a lower activation energy than β-scission." it is necessary to cite the authors.

 Author reply: We would like to thank you for reviewing our paper, we appreciate your insightful comments on our research.  The paper is cited:  Corma, A.; Planelles, J.; Sanchez-Marin, J.; Tomas, F., The role of different types of acid site in the cracking of alkanes on zeolite catalysts. Journal of Catalysis 1985, 93, (1), 30-37, doi.:10.1016/0021-9517(85)90148-4.

  1. Reduce the size of Figs. and letters in Figs. 3, 4 and 5 and scheme 1 in relation to other Figs. On page 11 and in key words, define the meaning of the abbreviation FCC.

  Author reply: We would like to thank you for reviewing our paper, we appreciate your insightful comments on our research. The size of Figs. and letters are reduced.  The abbreviation FCC is fluid catalytic cracking, which has been defined in paper.

  1. You have attached a supplementary file with the manuscript, which is actually the same as the original manuscript. Also, what is the purpose of the document “not published material” - do you consider it supplementary or not? I ask because you do not refer to the supplementary file in the text.

  Author reply: We would like to thank you for reviewing our paper, we appreciate your insightful comments on our research. The supplementary file is the color-labeled correction of the revised paper.

Reviewer 2 Report

Comments and Suggestions for Authors

ZSM-5 is one of the most studied and most discussed type of zeolitic materials in zeolite-based catalysis community. There have been numerous studies published analyzing its catalytic sites. In this regard although this paper is nicely written lack of significance and novelty. I would recommend some questions and corrections before its publication.

1.        Overall, this article lacks significance and novelty, there has been several papers some of them are seminal in field to discuss, I can name a few but most important chemical reviews by Mark Davis et al. need to cite and discussed. Authors need a suitable justification for why this new finding warrants publication and how novel it is.

2.       More experimental details needed, how the ZSMs are characterized.?

3.       Please provide labels and scale bar to SEM images (Fig 2)

4.       I have some concerns about the MAS NMRs, please provide full spectra in supplemental, how the difference is attributed to.

5.       Has the author done amine reduction to characterize the active sites, overall, some key characterizations of the zeolites are missing, please verify?

6.       How the zeolites behave in cyclic nature mean after multiple iterations of cracking, any studies done?

7.       Provide a conclusion and limitation of the studies.

Author Response

Point-to-point response

Dear Editors and Reviewers:

Thank you for your letter and the reviewers’ comments concerning our manuscript. Those comments are all very helpful for revising and improving our paper, as well as the important guiding significance to our research. We have studied the comments carefully and have made some changes to the manuscript. The changes are marked in yellow in the revised paper.

The responses of the reviewer’s comments and corrections in the paper are as follows:

Reviewer 1:

ZSM-5 is one of the most studied and most discussed type of zeolitic materials in zeolite-based catalysis community. There have been numerous studies published analyzing its catalytic sites. In this regard although this paper is nicely written lack of significance and novelty. I would recommend some questions and corrections before its publication.

  1. Overall, this article lacks significance and novelty, there has been several papers some of them are seminal in field to discuss, I can name a few but most important chemical reviews by Mark Davis et al. need to cite and discussed. Authors need a suitable justification for why this new finding warrants publication and how novel it is.

Author reply: We would like to thank you for reviewing our paper, we appreciate your insightful comments on our research. In the years 2020, 2021, and 2022, through applying two-dimensional 27Al correlation techniques coupled with series-connected hybrid (SCH) magnet at 35.2 T, J. L. White and his coworkers revealed that acidic ZSM-5 catalysts possess a minimum of two framework Al(IV) sites bonded to hydroxyl groups. The structural model of Al(IV)-II acid sites is still not clear. We proposed the structural model of Al(IV)-II acid sites in this work.

The most important chemical reviews by Mark Davis et al. have been cited and discussed in revised version.

ZSM-5 possesses acidic sites (Lewis acid sites and Brønsted acid sites) associated with its framework aluminum atoms. Davis and his coworkers have conducted extensive research to understand the influence of factors such as the Si/Al ratio, crystallization conditions, and post-synthesis treatments on acid site properties.[12-17] This understanding is essential for enhancing catalyst performance through optimization.

  1. More experimental details needed, how the ZSMs are characterized.?

Author reply: We would like to thank you for reviewing our paper, we appreciate your insightful comments on our research. In this work, X-ray Diffraction (XRD) is used to determine the crystal structure and phase purity of ZSM materials. Scanning Electron Microscopy (SEM) provides high-resolution images of the surface morphology and particle size distribution of ZSM materials. 1H and 27Al Solid-State Nuclear Magnetic Resonance (ssNMR) provides information about the local environment and chemical bonding of atoms within the ZSM framework.

  1. Please provide labels and scale bar to SEM images (Fig 2)

Author reply: We would like to thank you for reviewing our paper, we appreciate your insightful comments on our research. Labels and scale bars have been put on SEM images.

  1. I have some concerns about the MAS NMRs, please provide full spectra in supplemental, how the difference is attributed to.

Author reply: We would like to thank you for reviewing our paper, we appreciate your insightful comments on our research. Full spectra are provided in revised supplement.

  1. Has the author done amine reduction to characterize the active sites, overall, some key characterizations of the zeolites are missing, please verify?

    Author reply: We would like to thank you for reviewing our paper, we appreciate your insightful comments on our research. No amine reduction has been done to characterize the active sites. This work will be performed in our future work.

  1. How the zeolites behave in cyclic nature mean after multiple iterations of cracking, any studies done?

Author reply: We would like to thank you for reviewing our paper, we appreciate your insightful comments on our research. After hundreds of cyclic reactions, the zeolites lost their catalytic activities for coking.

  1. Provide a conclusion and limitation of the studies.

Author reply: We would like to thank you for reviewing our paper, we appreciate your insightful comments on our research.

Conclusion: In summary, NMR analyses demonstrate the presence of at least two distinct types of tetrahedral aluminum atoms within the zeolite framework. This suggests the existence of two chemically distinct Brønsted sites. The distribution of two chemically distinct Brønsted sites can be manipulated through appropriate adjustments to the synthesis conditions.

Limitations of the studies: Multiple analytical methods, such as TEM, IR, and X-ray Absorption Near Edge Structure, should be integrated to provide a more comprehensive understanding of ZSM-5 catalysts and their catalytic properties.

Reviewer 3 Report

Comments and Suggestions for Authors

In this study, ZSM-5 zeolites with different Al(IV) acid sites have been synthesized with or without TPA+ cations. The structures of hydroxyl groups in ZSM-5 zeolites were thoroughly investigated using 2D D-HMQC and 2D 1H MAS NMR experiments. This study provides valuable information on the quantity, strength, and location of acid sites. However, this paper also lack of some necessary explanation and discussion in determining the acid site. The followings are some questions.

1. In Figure 2, the author states that the signals between 1.2 and 2.2 ppm point to the presence of SiOH groups in 1H MAS NMR spectra. And much difference has been observed between three samples, what causes these results? More discussion should be added in this part.

2. Both ZSM-5 (high silica) and ZSM-5 (nano) in this study are proved to only have Al(IV)-1 acid sites based on the 27Al MAS NMR spectra. But the spectra shown in Figure 2 look very similar, How the authors come to this conclusion? Besides, 7 ppm in 1H MAS NMR spectra is assigned to Al(IV)-2 sites, there are also peaks belong to 7 ppm in both ZSM-5 (high silica) and ZSM-5 (nano), how to explain this peaks, and what do these peaks attribute to?

3. This paper did not show the physicochemical properties of reported samples, especially the acidic property. Since these three samples have totally different Si/Al ratio and thus different Brønsted acidic strength, which will influence the cracking of n-dodecane, how the authors dismiss this effect?  

4. Figure 7 shows the octane number of products created by three ZSM-5 catalysts, but the results show almost no difference, What are the advantages or particularities of ZSM-5 catalyst with predominant Al(IV)-2 acid sites?

Comments on the Quality of English Language

Nothing

Author Response

Point-to-point response

Dear Editors and Reviewers:

Thank you for your letter and the reviewers’ comments concerning our manuscript. Those comments are all very helpful for revising and improving our paper, as well as the important guiding significance to our research. We have studied the comments carefully and have made some changes to the manuscript. The changes are marked in yellow in the revised paper.

The responses of the reviewer’s comments and corrections in the paper are as follows:

Reviewer 2:

In this study, ZSM-5 zeolites with different Al(IV) acid sites have been synthesized with or without TPA+ cations. The structures of hydroxyl groups in ZSM-5 zeolites were thoroughly investigated using 2D D-HMQC and 2D 1H MAS NMR experiments. This study provides valuable information on the quantity, strength, and location of acid sites. However, this paper also lack of some necessary explanation and discussion in determining the acid site. The followings are some questions.

  1. In Figure 2, the author states that the signals between 1.2 and 2.2 ppm point to the presence of SiOH groups in 1H MAS NMR spectra. And much difference has been observed between three samples, what causes these results? More discussion should be added in this part.

Author reply: We would like to thank you for reviewing our paper, we appreciate your insightful comments on our research. In ZSM-5 zeolites, hydroxide groups can be present as part of the framework structure or as surface species.  The signals between 1.2 and 2.2 ppm are corresponding to surface Si-OH. These surface hydroxyl groups may result from the hydrolysis of silanol groups (Si-OH) on the external surface or in the channels on the zeolite inner surface. The variation in chemical shifts may arise from their location either on the external surface or within the micropores of the zeolite. 

  1. Both ZSM-5 (high silica) and ZSM-5 (nano) in this study are proved to only have Al(IV)-1 acid sites based on the 27Al MAS NMR spectra. But the spectra shown in Figure 2 look very similar, How the authors come to this conclusion? Besides, 7 ppm in 1H MAS NMR spectra is assigned to Al(IV)-2 sites, there are also peaks belong to 7 ppm in both ZSM-5 (high silica) and ZSM-5 (nano), how to explain this peaks, and what do these peaks attribute to?

Author reply: We would like to thank you for reviewing our paper, we appreciate your insightful comments on our research. No  1H-27Al 2D HMQC NMR spectra have been observed for both ZSM-5 (high silica) and ZSM-5 (nano), which indicated that the spatial of Al and H for Al(IV)-I Brønsted acid sites are not close enough. Compared with Al(IV)-II Brønsted acid sites, the framework Al(IV)-1 Brønsted acid sites have different structural properties, in which Al(IV)-1 is coordinated by four Si to form Al-(OSi)4.

  1. This paper did not show the physicochemical properties of reported samples, especially the acidic properties. Since these three samples have totally different Si/Al ratios and thus different Brønsted acidic strength, which will influence the cracking of n-dodecane, how the authors dismiss this effect?  

Author reply: We would like to thank you for reviewing our paper, we appreciate your insightful comments on our research. It is well known that the strength and density of the acid sites in ZSM-5 can indeed be adjusted or tuned through various synthesis conditions. For example, the Si/Al ratio in the synthesis gel affects the acidity of ZSM-5; and the choice and concentration of mineralizing agents (e.g., sodium hydroxide, tetrapropylammonium hydroxide) in the synthesis gel affect the pH and composition of the gel, which in turn influence the formation of ZSM-5 crystals and the acidity of the resulting material. Therefore, in our work, three different synthesis methods have been used to prepare ZSM-5 zeolites for identifying the acid sites in ZSM-5. Two of them only have Al(IV)-1 acid sites. The synthesis results indicate that the distribution of aluminum in zeolites can be manipulated through appropriate adjustments to the synthesis conditions. Examining the Brønsted acidic strength of ZSM-5 zeolites can be done through various experimental techniques and theoretical calculations. For example, TPD experiments involve the desorption of probe molecules, such as ammonia or pyridine, from the surface of the zeolite as a function of temperature;  IR spectroscopy can be used to monitor the vibrational frequencies of adsorbed probe molecules, such as pyridine or CO, on the surface of ZSM-5. However, our lab cannot perform those experiments due to the limitations of instruments.

  1. Figure 7 shows the octane number of products created by three ZSM-5 catalysts, but the results show almost no difference, What are the advantages or particularities of ZSM-5 catalyst with predominant Al(IV)-2 acid sites?

Author reply: We would like to thank you for reviewing our paper, we appreciate your insightful comments on our research. We did not observe any advantages of ZSM-5 catalyst with predominant Al(IV)-2 acid sites as catalysts. We will do much more work to prepare more Si/Al ratio samples for a deeper understanding of the catalytic properties of ZSM-5 catalyst with predominant Al(IV)-2 acid sites.

Round 2

Reviewer 1 Report

Comments and Suggestions for Authors

PREVIOUS COMMENT

  1. In the section Results and discussion in the text and Figures, it is necessary to clearly use the labels of the three zeolites. If you are in section 2.1. labeled the synthesized zeolites as Z1-Z3, then use these labels throughout the text. Basically, use the same labels throughout the manuscript.

NEW COMMENT

Please double-check the labels throughout the manuscript.

PREVIOUS COMMENT

  1. Clearly compare the characterization of all three zeolites in the text (Figs 1, 2, 3, 6). For example, in Figure 1 the peak intensities are not the same, the SEM image for Z3 is completely different from Z1 and Z2. Apply the same for the other images, and clearly indicate the observed differences using instrumental techniques for all three samples. Explanations were mostly explained collectively for all three samples.

Author reply: We would like to thank you for reviewing our paper, we appreciate your insightful comments on our research. In this work, three popular hydrothermal synthesis methods have been employed to prepare ZSM-5 zeolite. The acid sites in ZSM-5 are primarily attributed to the presence of Bronsted and Lewis acid sites. Bronsted Acid Sites are formed by the substitution of aluminum atoms with silicon atoms in the zeolite framework. This substitution creates a deficiency of electrons, leading to the formation of positively charged sites. These sites can donate protons (H+) and are responsible for catalyzing various acid-catalyzed reactions such as cracking, isomerization, and alkylation. Lewis Acid Sites are formed by the presence of aluminum atoms with incomplete coordination. These aluminum atoms have unoccupied orbitals capable of accepting electron pairs from Lewis bases. Lewis acid sites are also involved in various catalytic reactions, particularly in the activation of molecules like alkenes and aromatics. The distribution and strength of these acid sites in ZSM-5 can be influenced by factors such as the Si/Al ratio, crystal size, and synthesis conditions. Control over these factors allows for tailoring the acidity of ZSM-5 to suit specific catalytic applications, such as petroleum refining. In this work, the cracking of n-dodecane was performed to determine the acidic strength of two acid sites and evaluate the acid properties of ZSM-5 zeolites. However, no large difference is observed. The cracking reactions in ZSM-5 zeolites are minimally influenced by variations in the Si/Al ratio and crystal sizes.

NEW COMMENT:

There are no corrections in the text. It has not been corrected or explained why it was not corrected. Your comment is not in line with the suggestions.

PREVIOUS COMMENT

  1. The importance of the Z1 sample as a catalyst in relation to Z2 and Z3 should be better highlighted if possible in relation to the results of Figure 7.

Author reply: We would like to thank you for reviewing our paper, we appreciate your insightful comments on our research. The cracking reactions with different ZSM-5 zeolites are minimally influenced by variations in the Si/Al ratio, crystal sizes, and acid sites. Additional catalytic reactions should be conducted to differentiate between the catalytic activities of the two distinct acid sites.

NEW COMMENT:

There are no corrections in the text and additional explanations.

PREVIOUS COMMENT

  1. Figure 7 is very succinctly explained. The goal is to provide a solution. The goal is to provide answers to the question of which of these three samples is the most optimal. Will you get the answer from Figure 7, by comparing the results of instrumental techniques, cost of synthesis, lifetime of the catalyst, etc... The conclusion is incomplete. Complete the conclusion in relation to the previous comment.

Author reply: We would like to thank you for reviewing our paper, we appreciate your insightful comments on our research. Our research objective is to determine if the catalytic performance of Al(IV)-2 acid sites surpasses that of Al(IV)-1 acid sites. However, experimental results demonstrate that the catalytic performance of Al(IV)-2 acid sites is comparable with that of Al(IV)-1 acid sites. The conclusion is rewritten.

NEW COMMENT:

There are no corrections in the text and additional explanations.

PREVIOUS COMMENT

  1. Reduce the size of Figs. and letters in Figs. 3, 4 and 5 and scheme 1 in relation to other Figs. On page 11 and in key words, define the meaning of the abbreviation FCC.

Author reply: We would like to thank you for reviewing our paper, we appreciate your insightful comments on our research. The size of Figs. and letters are reduced. The abbreviation FCC is fluid catalytic cracking, which has been defined in paper.

NEW COMMENT:

„The abbreviation FCC is fluid catalytic cracking, which has been defined in paper.“ - Define in the correct place where the abbreviation first appears.

NEW COMMENT:

Complete the conclusions in accordance with the corrections.

Author Response

Point-to-point response

Dear Editors and Reviewers:

Thank you for your letter and the reviewers’ comments concerning our manuscript. Those comments are all very helpful for revising and improving our paper, as well as the important guiding significance to our research. We have studied the comments carefully and have made some changes to the manuscript. The changes are marked in yellow in the revised paper.

The responses to the reviewer’s comments and corrections in the paper are as follows:

 Reviewer 3:

  1. In the section Results and discussion in the text and Figures, it is necessary to clearly use the labels of the three zeolites. If you are in section 2.1. labeled the synthesized zeolites as Z1-Z3, then use these labels throughout the text. Basically, use the same labels throughout the manuscript.

NEW COMMENT

Please double-check the labels throughout the manuscript.

Author reply: We would like to thank you for reviewing our paper. After carefully checking, all labels (Z-I, Z-II, and Z-III) are kept throughout the manuscript.

  1. PREVIOUS COMMENT

Clearly compare the characterization of all three zeolites in the text (Figs 1, 2, 3, 6). For example, in Figure 1 the peak intensities are not the same, the SEM image for Z3 is completely different from Z1 and Z2. Apply the same for the other images, and clearly indicate the observed differences using instrumental techniques for all three samples. Explanations were mostly explained collectively for all three samples.

Author reply: We would like to thank you for reviewing our paper, we appreciate your insightful comments on our research. In this work, three popular hydrothermal synthesis methods have been employed to prepare ZSM-5 zeolite. The acid sites in ZSM-5 are primarily attributed to the presence of Bronsted and Lewis acid sites. Bronsted Acid Sites are formed by the substitution of aluminum atoms with silicon atoms in the zeolite framework. This substitution creates a deficiency of electrons, leading to the formation of positively charged sites. These sites can donate protons (H+) and are responsible for catalyzing various acid-catalyzed reactions such as cracking, isomerization, and alkylation. Lewis Acid Sites are formed by the presence of aluminum atoms with incomplete coordination. These aluminum atoms have unoccupied orbitals capable of accepting electron pairs from Lewis bases. Lewis acid sites are also involved in various catalytic reactions, particularly in the activation of molecules like alkenes and aromatics. The distribution and strength of these acid sites in ZSM-5 can be influenced by factors such as the Si/Al ratio, crystal size, and synthesis conditions. Control over these factors allows for tailoring the acidity of ZSM-5 to suit specific catalytic applications, such as petroleum refining. In this work, the cracking of n-dodecane was performed to determine the acidic strength of two acid sites and evaluate the acid properties of ZSM-5 zeolites. However, no large difference is observed. The cracking reactions in ZSM-5 zeolites are minimally influenced by variations in the Si/Al ratio and crystal sizes.

NEW COMMENT:

There are no corrections in the text. It has not been corrected or explained why it was not corrected. Your comment is not in line with the suggestions.

Author reply: We would like to thank you for reviewing our paper. The composition of the synthesis solution, including the types and concentrations of alkali metals, fluoride ions, organic structure-directing agents (OSDAs), or seeding agents, can significantly impact the morphology and properties of ZSM-5. In this work, three popular hydrothermal synthesis methods have been employed to prepare ZSM-5 zeolite, and XRD patterns and morphologies of ZSM-5 reflect the synthesized conditions. Corrections are in the revised paper.

  1. PREVIOUS COMMENT

The importance of the Z1 sample as a catalyst in relation to Z2 and Z3 should be better highlighted if possible in relation to the results of Figure 7.

Author reply: We would like to thank you for reviewing our paper, we appreciate your insightful comments on our research. The cracking reactions with different ZSM-5 zeolites are minimally influenced by variations in the Si/Al ratio, crystal sizes, and acid sites. Additional catalytic reactions should be conducted to differentiate between the catalytic activities of the two distinct acid sites.

NEW COMMENT:

There are no corrections in the text and additional explanations.

Author reply: We would like to thank you for reviewing our paper. The acidity of ZSM-5 catalysts, primarily due to the presence of Brønsted and Lewis acid sites, plays a crucial role in catalytic reactions. Variations in acidity can impact catalytic activity and selectivity in acid-catalyzed reactions. Corrections are in the revised paper.

  1. PREVIOUS COMMENT

Figure 7 is very succinctly explained. The goal is to provide a solution. The goal is to provide answers to the question of which of these three samples is the most optimal. Will you get the answer from Figure 7, by comparing the results of instrumental techniques, cost of synthesis, lifetime of the catalyst, etc... The conclusion is incomplete. Complete the conclusion in relation to the previous comment.

Author reply: We would like to thank you for reviewing our paper, we appreciate your insightful comments on our research. Our research objective is to determine if the catalytic performance of Al(IV)-2 acid sites surpasses that of Al(IV)-1 acid sites. However, experimental results demonstrate that the catalytic performance of Al(IV)-2 acid sites is comparable with that of Al(IV)-1 acid sites. The conclusion is rewritten.

NEW COMMENT:

There are no corrections in the text and additional explanations.

Author reply: We would like to thank you for reviewing our paper. The acidity of ZSM-5 catalysts, primarily due to the presence of Brønsted and Lewis acid sites, plays a crucial role in catalytic reactions. Variations in acidity can impact catalytic activity and selectivity in acid-catalyzed reactions. Brønsted acid sites are characterized by their ability to donate protons to reactant molecules, facilitating acid-catalyzed reactions such as cracking, isomerization, alkylation, and dehydration. The catalytic products suggest that the Brønsted acidic strength of the Al(IV)-2 sites is either a little stronger than or comparable to that of the Al(IV)-1 sites. Lewis acid sites are often associated with aluminum atoms in distorted tetrahedral positions or extra-framework cations. These sites exhibit a strong affinity for electron-rich species and can catalyze reactions involving Lewis acid-base interactions, such as adsorption, activation, and transformation of unsaturated molecules (e.g., olefins, aromatics). Lewis acid sites are essential for catalyzing reactions like alkene oligomerization, alkyl transfer, and aromatic substitution. Based on the catalytic findings, it is evident that both Z-II and Z-III zeolites possess a greater abundance of Lewis acid sites compared to Z-I zeolites. This abundance enhances their ability to catalyze isomerization reactions, resulting in a higher concentration of iso-paraffins in the liquid product. Corrections are in the revised paper.

  1. PREVIOUS COMMENT

Reduce the size of Figs. and letters in Figs. 3, 4 and 5 and scheme 1 in relation to other Figs. On page 11 and in key words, define the meaning of the abbreviation FCC.

Author reply: We would like to thank you for reviewing our paper, we appreciate your insightful comments on our research. The size of Figs. and letters are reduced. The abbreviation FCC is fluid catalytic cracking, which has been defined in paper.

NEW COMMENT:

„The abbreviation FCC is fluid catalytic cracking, which has been defined in paper.“ - Define in the correct place where the abbreviation first appears.

Author reply: We would like to thank you for reviewing our paper. Corrections are in the revised paper.

  1. NEW COMMENT:

Complete the conclusions in accordance with the corrections.

Author reply: We would like to thank you for reviewing our paper. Corrections are in the revised paper.

Reviewer 2 Report

Comments and Suggestions for Authors

fine to publish

Author Response

Dear Reviewers:

Thank you for your comments. 

Reviewer 3 Report

Comments and Suggestions for Authors

The MS has almost been revised based on  the comments. 

Author Response

(The authors gave the same response as above.)

Round 3

Reviewer 1 Report

Comments and Suggestions for Authors

The authors made some changes in the text. The XRD analysis is not explained in the way I suggested, and neither are the results in Figure 7. However, I will accept the manuscript in this form and let the Editor make the final decision.

Author Response

Thank you for yourcomments concerning our manuscript. Those comments are very helpful for revising and improving our paper and have important guiding significance to our research.